# Lateral Diffusion of NKCC1 Contributes to Chloride Homeostasis in Neurons and Is Rapidly Regulated by the WNK Signaling Pathway

**DOI:** 10.3390/cells12030464

**Published:** 2023-01-31

**Authors:** Etienne Côme, Simon Blachier, Juliette Gouhier, Marion Russeau, Sabine Lévi

**Affiliations:** INSERM UMR-S 1270, Institut du Fer à Moulin, Sorbonne Université, 75005 Paris, France

**Keywords:** hippocampal neurons, chloride homeostasis, GABAergic transmission, lateral diffusion, clustering, protein trafficking, signaling, single molecule localization microscopy

## Abstract

An upregulation of the Na^+^-K^+^-2Cl^−^ cotransporter NKCC1, the main chloride importer in mature neurons, can lead to depolarizing/excitatory responses mediated by GABA type A receptors (GABA_A_Rs) and, thus, to hyperactivity. Understanding the regulatory mechanisms of NKCC1 would help prevent intra-neuronal chloride accumulation that occurs in pathologies with defective inhibition. The cell mechanisms regulating NKCC1 are poorly understood. Here, we report in mature hippocampal neurons that GABAergic activity controls the membrane diffusion and clustering of NKCC1 via the chloride-sensitive WNK lysine deficient protein kinase 1 (WNK1) and the downstream Ste20 Pro-line Asparagine Rich Kinase (SPAK) kinase that directly phosphorylates NKCC1 on key threonine residues. At rest, this signaling pathway has little effect on intracellular Cl^−^ concentration, but it participates in the elevation of intraneuronal Cl^−^ concentration in hyperactivity conditions associated with an up-regulation of NKCC1. The fact that the main chloride exporter, the K^+^-Cl^−^ cotransporter KCC2, is also regulated in mature neurons by the WNK1 pathway indicates that this pathway will be a target of choice in the pathology.

## 1. Introduction

Upon GABA activation, GABA type A receptors (GABA_A_Rs) activate a selective chloride/bicarbonate conductance in neurons. The direction of chloride (Cl^−^) flux through the channel depends on transmembrane Cl^−^ gradients. Therefore, Cl^−^ homeostasis determines the efficacy of GABAergic transmission. Drug-resistant epilepsies are often associated with defective Cl^−^ transport [1]. It is, therefore, crucial to find new mechanisms regulating neuronal Cl^−^ transport that could eventually lead to the development of treatments for drug-resistant epilepsies and diseases with decreased inhibition, such as neuropathies and neuropsychiatric disorders [2].

The increase in [Cl^−^]_i_ and the consequent impact on the GABA reversal potential (E_GABA_) in various models of epilepsy is often attributed to a malfunction of the neuronal K^+^-Cl^−^ cotransporter KCC2, which exports Cl^−^ from neurons [3]. In addition, increased expression and/or activity of the Na^+^-K^+^-Cl^−^ cotransporter NKCC1, which imports chloride into neurons, also contributes to increased [Cl^−^]_i_ and depolarization of E_GABA_. This has been observed in the subiculum of temporal lobe epilepsy (TLE) patients [4,5] as well as in several in vivo and in vitro epilepsy models, such as those induced by a glioma [6] or traumatic brain injury [7] models and the kainic acid and pilocarpine epilepsy models in tissue slices [8,9,10]. Targeting NKCC1 in epilepsy with the inhibitor bumetanide is protective: it abolished glioma-induced seizures in rats [6] and reduced the frequency of interictal-like activities [4], the duration of ictal activities [10], and the sprouting of mossy fibers [9].

Key cellular and molecular mechanisms regulate the membrane trafficking of KCC2. They involve activity-dependent control mechanisms [11,12]. An increase in excitatory activity regulates KCC2 cell trafficking in mature neurons within minutes [13,14,15]. N-methyl-D-aspartate receptor (NMDAR)-induced Ca^2+^-influx activates protein phosphatase 1 (PP1) dephosphorylation of KCC2 serine S940 (S940) and cleavage of its C-terminal domain by Ca^2+^-activated calpain [14,16,17]. This increases the lateral diffusion, endocytosis, and degradation of KCC2 [13,14]. GABAergic signaling also tunes KCC2 at the neuronal membrane through GABA_A_R activity and Cl^−^-dependent phosphorylation of KCC2 Threonine 906 and 1007 (T906/1007) residues [18]. The second messenger Cl^−^ controls the activity of WNK lysine deficient protein kinase 1 (WNK1), a serine-threonine kinase that senses Cl^−^, and regulates Ste20 Pro-line Asparagine Rich Kinase (SPAK) and Oxidative Stress Response kinase 1 (OSR1) [19].

Interestingly, this signaling not only promotes the phosphorylation of KCC2 threonine T906/T1007 but also of NKCC1 threonine T203/T207/T212 [20]. The resulting inhibition of KCC2 and activation of NKCC1 contribute to elevating [Cl^−^]_i_ in non-neuronal cells [21]. Therefore, blocking the phosphorylation of KCC2 and NKCC1 via WNK signaling would prevent this deregulation of the transporters, thus preventing excessive accumulation of Cl^−^ in neurons in pathological conditions.

Our team has shown the contribution of surface diffusion in the rapid control of KCC2 membrane stability and Cl^−^ neuronal homeostasis in response to changes in neuronal activity [13,18]. The transporter alternates between periods of confinement within clusters near synapses and periods of free movement outside the clusters. The transporter that freely diffuses into the membrane may then encounter clathrin-coated pits and be retained there for internalization and then recycled back to the membrane by exocytosis or degraded. The pools of free vs. confined transporters are in dynamic equilibrium, allowing fine-tuning of chloride levels at, e.g., synapses following fluctuations in synaptic activity [11,12]. As regulations of KCC2 mobility occur within seconds [18], membrane dynamics is probably the main cell mechanism controlling KCC2 trafficking at the plasma membrane.

Recently, we showed that NKCC1 is targeted to axons and somato-dendritic compartments and that it forms clusters at the periphery of synapses of hippocampal neurons [12]. Single particle tracking using quantum dots (QD-SPT) in living neurons indicated that in the somato-dendritic compartment, NKCC1 transporters scan large areas of the extrasynaptic membrane while others explore smaller areas and are confined, e.g., near synapses [12]. The transitions of NKCC1 between synaptic and extrasynaptic areas are reminiscent of the diffusion of KCC2 [13,18], suggesting that NKCC1 responds to the “diffusion-trap” mechanism just like KCC2 [12]. A rapid, activity-dependent regulation of NKCC1 diffusion/clustering may therefore affect its function and, consequently, intracellular chloride levels and GABA signaling.

Here we show that neuronal GABAergic activity rapidly regulates surface diffusion and clustering of NKCC1 in mature hippocampal neurons. Blocking or activating GABA_A_ receptors with muscimol or gabazine confines the transporters to the dendritic membrane. GABAergic activity would control diffusion and clustering of NKCC1 in the dendrite via the chloride-sensitive kinase WNK1 and the kinases SPAK and OSR1 that would directly phosphorylate NKCC1 at key threonine phosphorylation sites T203/T207/T212. Our results indicate that this regulation would be particularly effective in regulating NKCC1 membrane diffusion, clustering, and function, and thereby chloride homeostasis in the somato-dendrtic compartment under conditions of hyperactivity, highlighting the interest of targeting the WNK1/SPAK/OSR1 pathway in pathologies with inhibition defect.

## 2. Materials and Methods

The animal procedures for the experiments performed on primary cultures of hippocampal neurons were carried out in accordance with the directive of the Council of the European Community of 24 November 1986 (86/609/EEC) and the directives of the French Ministry of Agriculture and the Direction Départementale de la Protection des Populations de Paris (Institut du Fer à Moulin, Animalerie des Rongeurs, license C 72-05-22). Animal suffering and the number of animals were reduced to a minimum. Pregnant Sprague-Dawley rats were obtained from Janvier, and the embryos of either sex were used at embryonic day 18 or 19.

### 2.1. Neuronal Culture

Primary cultures of hippocampal neurons were prepared as previously described [13]. Briefly, dissected hippocampi were trypsinized (0.25% *v*/*v,* Sigma-Aldrich, Lyon, France) and mechanically dissociated in 1 × Hanks’ Balanced Salt solution (HBSS, Invitrogen, Cergy Pontoise, France) containing 10 mM 2-Hydroxyethyl)piperazine-N′-(2-ethanesulfonic acid (HEPES, Invitrogen). Neurons were seeded (density = 120 × 10^3^ cells/mL) on glass coverslips (Assistent, Winigor, Germany) that were pre-coated overnight with poly-D,L Ornithine (50–75 µg/mL, Sigma) in H_2_O. Neurons were allowed to attach to the coverslips in the presence of horse serum (10% *v*/*v*, Invitrogen) in Minimum Essential Medium (MEM, Sigma) supplemented with L-glutamine (2 mM, Invitrogen) and Na^+^ pyruvate (1 mM, Invitrogen). Cells were incubated at 37 °C in a humidified 5% CO_2_ incubator. After 3–4 h, the attachment medium was removed and replaced with Neurobasal medium supplemented with B27 (1x), L-glutamine (2 mM), and antibiotics (penicillin 200 units/mL, streptomycin, 200 µg/mL, Invitrogen). Cultures were kept for 3–4 weeks at 37 °C in a 5% CO_2_ incubator. Each week, the culture medium was renewed (one-fifth of the volume was added).

### 2.2. DNA Constructs

The pcDNA3.1 Flag YFP hNKCC1 HA-ECL2 (NT931) was obtained from Biff Forbush (Addgene plasmid # 4906; RRID:Addgene_49063; Addgene, Watertown, MA, USA) [22]. From this NKCC1-HA-Flag-mVenus plasmid, the following constructs were raised: NKCC1-HA-Δflag-ΔmVenus by truncation of the tags located on NKCC1 NTD, NKCC1-TA3-Flag-mVenus, and NKCC1-TA3-ΔFlag-ΔmVenus with mutation of T203/207/212 to Alanines, and NKCC1-TA5-Flag-mVenus and NKCC1-TA5-ΔFlag-ΔmVenus with mutation of T203/207/212/217/230 to alanines. The threonine nucleotide sequence was changed to GCA for alanine substitution. The following constructs were also used: pCAG_rat KCC2-3Flag-ECL2 [13], pCAG_KCC2-3Flag-ECL2 T906/1007E [18], eGFP (Clontech, Saint-Germain-en-Laye, France), pCAG_GPHN.FingR-eGFP-CCR5TC [23] (gift from Don Arnold, Addgene plasmid # 46296; RRID:Addgene_46296; Addgene), homer1c-DsRed (kindly provided by D. Choquet, IIN, Bordeaux, France), WNK1 with “kinase-dead, dominant-negative domain” (WNK1-KD, D368A), “constitutively active” WNK1 (WNK1-CA, S382E) [24] (kindly provided by I. Medina, INMED, Marseille, France), and SuperClomeleon [25] (kindly provided by G.J. Augustine, NTU, Singapore). All constructs were sequenced by Beckman Coulter Genomics (Villepinte, France).

### 2.3. Neuronal Transfection

Neurons were transfected at 13–14 days in vitro (DIV) using Transfectin (BioRad, Marnes-la-Coquette, France) with a DNA:transfectin ratio of 1 µg:3 µL, with 1–2 µg of plasmid DNA per well. Simple transfections of NKCC1-HA-Flag-mVenus plasmid concentration: 1 µg. The following amount of plasmids was used: 1:0.4:0.4 µg for NKCC1 constructs together with GPHN.FingR-eGFP and homer1c-DsRed; 1:0.2 µg for NKCC1 constructs with eGFP; 0.7:0.7 µg for NKCC1 constructs with WNK1-KD or WNK1-CA, NKCC1 constructs with SuperClomeleon; 0.7:0.7 µg KCC2 constructs with SuperClomeleon; 0.5:0.5:0.5 µg SCLM + KCC2 + NKCC1. Experiments were carried out 7–10 days after transfection. QD-SPT, Stochastic Optical Reconstruction Microscopy (STORM), and chloride imaging experiments were performed with Δflag-ΔmVenus NKCC1 constructs. Standard epifluorescence microscopy with Flag-mVenus NKCC1 constructs.

### 2.4. Peptide Treatment and Pharmacology

Experiments were performed using the following drugs: Dynamin inhibitor (myristoylated peptide, 50 µM; Tocris Bioscience, Noyal Châtillon sur Seiche, France), TTX (1 µM; Latoxan, Valence, France), R,S-MCPG (500 µM; Abcam, Paris, France), S-MCPG (250 µM; HelloBio, Bristol, UK), Kynurenic acid (1 mM; Abcam), gabazine (10 µM; Abcam), muscimol (10 µM; Abcam), WNK463 (10 µM, MedChemTronica, Stockholm Sweden), closantel (10 µM; Sigma), bumetanide (5 µM, Abcam). R,S-MCPG and S-MCPG were dissolved in equimolar concentrations of NaOH; TTX was prepared in 2% citric acid (*v*/*v*); closantel was solubilized in DMSO (Sigma).

For QD-SPT experiments, coverslips were mounted in a recording chamber. Before imaging, cells were incubated with the appropriate drugs and/or peptide at 32 °C for 10 min in the imaging medium (see below). They were then imaged for 45 min in the presence of the drugs or peptide. For immunocytochemistry, drugs were directly added to the culture medium and incubated for 30 min in a CO_2_ incubator set at 37 °C before cell fixation. The imaging medium was composed of phenol red-free MEM supplemented with glucose (33 mM; Sigma), HEPES (20 mM), glutamine (2 mM), Na^+^-pyruvate (1 mM), and B27 (1x) (all from Invitrogen). The 138 mM [Cl^−^] solution consisted of CaCl_2_ (2 mM), KCl (2 mM), MgCl_2_ (3 mM), HEPES (10 mM), glucose (20 mM), NaCl (126 mM), and Na^+^ methane sulfonate (15 mM); the 0 mM [Cl^−^] solution was composed of CaSO_4_ (1 mM), K^+^-methane sulfonate (2 mM), MgSO_4_ (2 mM), HEPES (10 mM), glucose (20 mM), and Na^+^-methane sulfonate (144 mM).

### 2.5. Staining for Single-Particle Tracking

Neurons were incubated with primary antibodies against hemagglutinin (HA) (rabbit, 1:250, cell signaling technology, Leiden, The Netherlands, cat #C29F4) for 8 min at 37 °C. After several washes in the imaging medium (see composition above), cells were incubated for 30–60 s at 37 °C with F(ab’)_2_-goat anti-rabbit antibodies coupled to QDot at 655 nm (1 nM; Invitrogen) in phosphate buffer saline (PBS, 1x; Invitrogen) supplemented with Casein (10% *v*/*v*, Sigma).

### 2.6. QD-Based Single-Particle Tracking

Neurons were imaged at 33 °C with an inverted Olympus IX71 microscope, a 60X objective (Numerical Aperture NA 1.42; Olympus, Rungis, France), and a Mercury lamp (X-Cite 120Q, Lumen Dynamics, Mississauga, ON, Canada). The diffusion of NKCC1 along axons was studied on neurons co-transfected with NKCC1 and eGFP plasmids. eGFP labeling allowed for distinguishing dendrites from axons. Indeed, the eGFP staining uniformly fills the soma, dendrites, and axons of the neurons, allowing seeing the overall morphology of the neuron. Mature neurons have very thin and long axons that are easily distinguishable from much thicker and shorter dendrites. Moreover, since the transfection efficiency is very low, it is easy to see the origin of the axon of a given neuron. We, therefore, imaged the axons of isolated neurons in their first hundred micrometers. The diffusion of NKCC1 at synapses vs. at the distance of synapses was followed in neurons transfected with NKCC1 constructs together with GPHN.FingR-eGFP and homer1c-DsRed plasmids. Individual GPHN.FingR-eGFP and homer1c-DsRed images were acquired with an ImagEM Electron-multiplying CCDs (EMCCD) camera and MetaView software (Meta Imaging Series 7.8). QD real-time movies of 1200 frames were acquired with a 30 ms integration time. 

QD tracking and trajectory reconstruction was performed with homemade software (Matlab R2020A; The Mathworks, Natick, MA, USA) as described in [26]. Trajectories were considered synaptic when overlapping with the synaptic mask of GPHN.FingR-eGFP or homer1c-DsRed clusters, or extrasynaptic for spots four pixels (760 nm) away. The inclusion area was increased relatively to previous studies on KCC2 [13,18] due to the low numbers of NKCC1 trajectories recorded with a 380 nm distance. Expanding the radius did not actually change results for perisynaptic NKCC1 lateral diffusion, suggesting that its clusters are located further away from synapses than KCC2 ones. One to two dendritic and axonal regions were analyzed per cell. The QD crossing trajectories were removed from the analysis. 

The mean square displacement (MSD) vs. time curves were determined for each trajectory by using the equation:$$\mathrm{MSD}\left(n\tau \right)=\frac{1}{N-n}\overset{N-n}{\underset{i=1}{\sum }}\left[\left(x_{i+n}-x_{i}\right)²+\left(y_{i+n}-y_{i}\right)²\right]$$

In this equation, τ = acquisition time, N= total number of frames, i = positive integers, and *n*= time increment. Diffusion coefficients (D) were obtained by fitting the first four points without the origin of the MSD vs. time curves with MSD(nτ) = 4Dnτ + σ, with σ being the spot localization accuracy. The pointing accuracy of the QD is ~20–30 nm. The explored area corresponds to the MSD value of the trajectory at two different time intervals of 0.42 and 0.45 s [27]. The dwell time at synapses was defined as the duration of detection of QDs at synapses divided by the number of exits [26]. Dwell times lower than five frames were discarded from the analysis. 

### 2.7. Chloride Imaging

Imaging was performed at 33 °C on our Olympus microscope with a 60x (1.42 NA) objective. A lambda DG-4 monochromator (Sutter Instruments) was used as the light source and to shift rapidly between CFP and YFP filters. The following filters were used: excitation, D436/10X and HQ485/15X; dichroic, 505DCXR; emission, HQ510lp; Chroma Technology, Olching Germany). Image acquisition was performed with an ImagEM EMCCD camera (Hamamatsu Photonics, Massy, France) controlled by MetaFluor software Series 7.8.9.0 (Roper Scientific, Evry, France). Sixteen-bit images (512 × 512) were acquired every 30 s for 5 min (acquisition time of 30 ms). After background substraction, the F480/F440 ratio was measured. 

### 2.8. Immunocytochemistry

NKCC1-HA-Flag-mVenus membrane expression and clustering were assessed with staining performed after a 4 min fixation at room temperature (RT) in paraformaldehyde (PFA; 4% *w*/*v*; Sigma) and sucrose (20% *w*/*v*; Sigma) solution in PBS (1×). Cells were washed three times in PBS and incubated for 30 min at RT in a blocking solution composed of goat serum (GS; 3% *v*/*v*; Invitrogen) prepared in PBS. Neurons were then incubated for 60–180 min at RT with HA primary antibody (rabbit, 1:250, cell signaling technology, cat #C29F4) in PBS–GS blocking solution. After washing, neurons were incubated with Cy™3 AffiniPure Donkey Anti-Rabbit IgG (H + L) (1.9 µg/mL; Jackson ImmunoResearch, Saint-Cyr-L’École, France, cat #111-165-003) for standard epifluorescence assays, or Alexa Fluor^®^ 647 AffiniPure Donkey Anti-Rabbit IgG (H + L) (2 µg/mL, Jackson ImmunoResearch, cat #711-605-152) for super-resolution experiments, in PBS-GS solution. The coverslips were then washed and mounted on slides with mowiol 844 (48 mg/mL; Sigma). Sets of neurons compared for quantification were labeled and imaged simultaneously.

### 2.9. Fluorescence Image Acquisition and Analysis

Image acquisition was performed using a 100× objective (1.40 NA) on a Leica (Nussloch, Germany) DM6000 upright microscope with a 12-bit cooled CCD camera (Micromax, Roper Scientific) using MetaMorph software Series 7.8 (Roper Scientific). To assess NKCC1-HA clusters, the exposure time was determined on the brightest experimental condition in order to be non-saturating and was fixed for all cells and conditions to be analyzed. Quantification was performed using a MetaMorph routine (Roper Scientific). For the dendritic intensity and clustering analysis, a region of interest (ROI) was traced around a selected dendrite, and the average pixel intensity in the ROI was measured. For the clustering analysis, images were filtered using the flattened background (kernel size, 3 × 3 × 2) function on Metamorph to enhance cluster outlines, and an intensity threshold defined by the user was set to identify the clusters and avoid their coalescence. Clusters were outlined, and the corresponding regions were transferred onto the raw images to determine NKCC1-HA cluster number, surface, and fluorescence intensity. To estimate the membrane fraction of NKCC1, the mean pixel intensity of Venus and of Cy3-tagged membrane NKCC1-HA were computed from a dendritic ROI, and the surface/total ratio of intensity was calculated. The area of the dendritic region analyzed was used to calculate the number of clusters per surface area. We analyzed ~10 cells per experimental condition and per culture.

### 2.10. STORM Microscopy

STORM was performed on fixed samples on an inverted N-STORM Nikon Eclipse Ti microscope (Nikon, Lisses, France) equipped with a 100× objective (NA 1.49) and an Andor iXon Ultra EMCCD camera, and using specific lasers for STORM of Alexa 647. Image acquisition consisted of the accumulation of 30,000 frames with a 50 ms frame rate. A Nikon perfect focus system was used to maintain the z position during the whole acquisition. Single-molecules were localized, and 2D images were reconstructed as described [28] by fitting the PSF of fluorophores to a 2D Gaussian distribution. The position of fluorophores was corrected by the relative movement of the cluster by calculating the center of mass of the cluster throughout the acquisition using a partial reconstruction of 2000 frames with a sliding window [28]. Rendered images were obtained by superimposing the coordinates of single molecule detections. To correct multiple detections obtained from the same molecule of Alexa 647, detections occurring in the vicinity of space (2σ) and time (15 s) were identified as belonging to the same molecule. 

The number of NKCC1 clusters, their size, and the density of detections in the pixels of a cluster were measured on 2D images reconstructed through cluster segmentation based on detection densities. The minimal thresholds to determine clusters were 1% intensity, 0.1 per nm² minimum detection density, and 10 detections. The resulting binary image was analyzed with the function “regionprops” of Matlab to extract the surface area of each cluster identified by this function. The molecular density corresponds to the number of detections in the pixels (STORM pixel size = 20 nm) that belong to a cluster divided by the cluster area.

### 2.11. Statistics

N corresponds to the number of QDs for SPT experiments, clusters for STORM, and cells for immunocytochemsitry and chloride imaging. The sample size was determined from published work, pilot experiments, and our own expertise. Mostly all data were used. For SuperClomeleon imaging, SPT or immunocytochemsitry, suffering cells with blobs and/or fragmented neurites were discarded from the analysis. Data representation was performed with boxplots or cumulative frequency plots. The statistical test to compare the two groups was either the Welch *t*-test when the normality assumption was met (Q-Q plots and cumulative frequency fit); otherwise, the Mann–Whitney test was performed to assess the presence of dominance between the two distributions. For variables following a log-normal distribution, such as those obtained from SPT and STORM assays, we applied the log(.) function after dividing by the control group’s median. For super-resolution experiments, as an important variability could be observed between different cells in the same coverslip, a balanced random selection of clusters across neurons, conditions, and cultures was performed, then variables from each culture were divided by the median of the control group. Results from different cultures were pooled, and log(.) was applied, then the Mann–Whitney U value was computed. The process was repeated 1000 times, and the *p*-value was determined from the U distribution using the basic definition of the *p*-value. For SPT analysis, note that each QD is associated with 3 EA values; thus, the sample size is three times greater. Statistical analysis was performed with R (3.6.1, R Core Team, 2019, Vienna, Austria) using the following package (ggplot2, matrixStats). Differences were significant for *p*-values less than 5% (* *p* < 0.05; ** *p* < 0.01; *** *p* < 0.001; NS, not significant).

## 3. Results

### 3.1. Opposite Effects of GABAergic Activity on NKCC1 Mobility in the Axon vs. the Dendrite

We questioned whether inhibitory GABAergic transmission influences NKCC1 lateral diffusion in mature hippocampal cultures at 21–23 DIV using QD-SPT. We showed by Western blot in a previous study that mature (21 DIV-old) hippocampal cultures express endogenous NKCC1 [18]. The expression level of NKCC1 in our cultures does not differ from that in hippocampal tissue, suggesting that this expression is not an artifact of the culture [18]. Here, we determined the impact of a pharmacological activation or blockade of GABA_A_Rs in the presence of tetrodotoxin (TTX), kynurenate (KYN), and mCPG (MCPG) to block action potentials and glutamatergic activity and compared to the TTX + KYN + MCPG “control” condition. In the absence of antibodies targeting a specific extracellular epitope of NKCC1 at the surface of living neurons, we transfected hippocampal cultures with recombinant HA-tagged NKCC1 constructs at 14 DIV. Then, we stained them at 21–24 DIV with an anti-HA antibody coupled to QDs and imaged them (see Materials and Methods). First, we reported that in spontaneous activity conditions, NKCC1 diffuses along the axon (Appendix A) and in the somato-dendritic compartment (Figure 1A). Neurons were acutely exposed to the GABA_A_R agonist muscimol (10 μM) or competitive antagonist gabazine (10 μM), two drugs that were shown using electrophysiological recordings to increase or block GABA_A_R-mediated inhibition in mature hippocampal neurons [18]. We observed that, when exposed to gabazine or muscimol, QDs explored more the surface of the axon compared to QDs in control conditions (Appendix A). The slope of the mean square displacement (MSD) as a function of time was increased for trajectories recorded in gabazine and muscimol conditions compared to controls (Appendix A), indicating reduced confinement of NKCC1 in the axon. This was associated with an increase in the diffusion coefficient (Appendix A) and the explored area (Appendix A). Therefore, NKCC1 confinement is reduced in the axon under conditions of GABAergic activity blockade.

Moreover, we found that GABAergic signaling oppositely regulated the mobility of NKCC1 in the axon vs. the dendrite. In contrast to what we found in the axon, we showed that QDs explored smaller areas of the dendritic membrane following exposure to gabazine or muscimol compared to the control (Figure 1A). Analysis performed on the whole population (extrasynaptic + synaptic) of dendritic trajectories revealed that the MSD function displayed a less steep slope for trajectories recorded in the presence of muscimol or gabazine as compared with control (Figure 1B), indicative of increased confinement upon activation or blockade of the GABA_A_R. The median diffusion coefficient and explored area values of dendritic NKCC1 were also reduced upon muscimol and gabazine application (Figure 1C,D, respectively). Thus, the lateral diffusion of NKCC1 on the dendrite is regulated by inhibitory GABAergic transmission: the transporters are slowed down and confined in response to a change in GABAergic activity.

We then analyzed the effects of gabazine and muscimol on the diffusion of NKCC1 at extrasynaptic and synaptic sites. Muscimol and gabazine reduced the transporter mobility and surface exploration of individual QDs (Figure 1E). Quantitative analysis on populations of QDs revealed an impact of the treatments on diffusion coefficient and explored the area for extrasynaptic QDs and for QDs at both excitatory and inhibitory synapses (Figure 1F,G). The effect was greater on the diffusion coefficient than on the explored area for NKCC1 trajectories located at the periphery of glutamatergic synapses and vice versa for trajectories near inhibitory synapses (Figure 1F,G). Thus, NKCC1 exhibits increased diffusion constraints at extrasynaptic sites and at the periphery of synapses upon GABA_A_R activation or blockade.

### 3.2. NKCC1 Is Targeted to Endocytic Zones Where They Are Stored upon GABAergic Activity Changes

We have shown (unpublished work) that NKCC1, unlike KCC2, is confined within endocytic zones without necessarily being internalized. This suggests that NKCC1 can be stored in endocytic zones. This pool of NKCC1 would constitute a reserve of membrane transporters, which would be rapidly released in the plasma membrane under specific conditions. We observed that acute exposure to muscimol or gabazine increased the confinement of NKCC1 in endocytic zones, as observed in neurons transfected with clathrin-YFP for individual trajectories (Figure 1H) or for hundreds of molecules (Figure 1I,J). Moreover, blocking clathrin-mediated endocytosis with an inhibitory peptide prevented the slow down and increased confinement of NKCC1 upon gabazine or muscimol treatment (Figure 1K,L). We, therefore, concluded that the treatment of neurons with muscimol or gabazine increased the confinement of NKCC1 transporters in endocytic zones in the dendrites.

To investigate whether the increased confinement of the transporter to endocytic zones induced by GABA_A_R agonists and antagonists is accompanied by an increase in its internalization, we analyzed the surface pool of NKCC1 (measured as the ratio of the mean fluorescence intensity of the surface/surface + intracellular NKCC1) (Figure 2A,B). The ratio remained unchanged (Figure 2B) after exposure to muscimol or gabazine for 30 min, indicating that changes in GABA_A_R activity do not affect the membrane stability of NKCC1. This is consistent with the regulation of KCC2 diffusion by GABAergic inhibition, which regulates the transporter clustering and thus function without requiring transporter internalization [18]. We examined whether changes in GABA_A_R-dependent inhibition could alter NKCC1 clustering. Using conventional epifluorescence, we reported that muscimol significantly reduced by 1.28-fold the density of NKCC1 clusters at the surface of neurons transfected with NKCC1-HA (Figure 2C). Furthermore, a 30 min exposure to muscimol reduced by 1.1-fold the size of NKCC1 clusters (Figure 2D), as compared with untreated cells. In contrast, muscimol did not affect the fluorescence intensity of the clusters (Figure 2E). These results indicate that the increased confinement of NKCC1 in endocytic zones induced upon muscimol treatment correlates with a reduction in its membrane clustering. Unlike muscimol, gabazine did not noticeably alter the density of NKCC1 clusters (Figure 2C). However, it significantly reduced the size (Figure 2D) and intensity (Figure 2E) of these clusters, suggesting transporter loss within clusters.

Since NKCC1 cluster size is at the limit of the resolution of a standard epifluorescence microscope, we further analyzed the effect of the treatments on NKCC1 clustering using super-resolution STORM. NKCC1 forms round-shaped clusters along the dendrites (Figure 2F). We report that neuronal exposure to gabazine or muscimol altered the nanoscopic organization of NKCC1 (Figure 2F). The muscimol treatment significantly decreased by 1.16-fold, the average size of NKCC1 nanoclusters (Figure 2G). This effect was not accompanied by a decreased number of molecules per cluster (Figure 2H). In fact, the density of particles detected per cluster was significantly increased by 1.43-fold after muscimol exposure (Figure 2I), indicating molecular compaction. This effect, coupled with the loss of NKCC1 clusters observed with standard epifluorescence and the increased confinement of the transporter in endocytic zones, is consistent with a muscimol-induced escape of NKCC1 transporters from membrane clusters followed by their recruitment and storage in endocytic zones.

STORM microscopy revealed that gabazine treatment significantly decreased the size of NKCC1 clusters (by 0.62-fold, Figure 2G), accompanied by a 1.5-fold reduction in the number of molecules per cluster (Figure 2H), in agreement with the notion that transporters escaped clusters. This effect was not associated with a change in the density of molecules per cluster (Figure 2I), reporting no change in the compaction of molecules within the cluster. Although the effects of muscimol and gabazine differ on the nanoscale organization of NKCC1, both treatments lead to the escape of transporters from clusters, which are then rapidly captured in the endocytic zones where they are stored.

### 3.3. Intracellular Chloride Levels Tune NKCC1 Surface Diffusion and Clustering

We then investigated whether variations in [Cl^−^]_i_ could explain the effects of gabazine and muscimol treatment on NKCC1 diffusion, as shown for KCC2 [18]. We lowered [Cl^−^]_i_ by replacing Cl^−^ with methane sulfonate in the imaging medium and tested its effect on NKCC1 diffusion. We showed that in similar 21 DIV-old hippocampal neurons, lowering the extracellular chloride in the medium by replacing it with methane sulfonate significantly increases (+25%) the FRET ratio of the SuperClomeleon signal as expected for a decrease in [Cl^−^]_i_ [18]. The decrease in [Cl^−^]_i_ increased NKCC1 surface exploration for individual trajectories located at the periphery of synapses and at a distance (Figure 3A). Quantification revealed that this treatment had no effect on the diffusion coefficients of dendritic NKCC1 for either extrasynaptic or synaptic trajectories (Figure 3B). On the other hand, lowering [Cl^−^]_i_ decreased the confinement of extrasynaptic transporters (Figure 3C), while the confinement of synaptic transporters was unchanged (Figure 3C). We then asked if the reduced confinement observed at extrasynaptic sites concerned transporters stored in endocytic zones. We found that the diffusion coefficient and surface exploration of NKCC1 were significantly increased for transporters located at a distance of clathrin-coated pits (Figure 3D–F) while the mobility of transporters in endocytic zones was not modified upon lowering [Cl^−^]_i_ (Figure 3E,F). Therefore, reducing [Cl^−^]_i_ does not increase the confinement of NKCC1 in endocytic zones. Altogether, our results provide evidence that lowering intracellular chloride levels removes diffusion constraints onto NKCC1, which moves faster in the membrane, probably by being relieved from endocytic zones.

We then determined whether this relief in diffusion constraints of the transporter was associated with its membrane redistribution. Quantification of the membrane pool of NKCC1 revealed that lowering [Cl^−^]_i_ increased NKCC1 immunoreactivity on the dendrites (Figure 3G). This was correlated with a 1.25-fold increase in the surface amount of NKCC1 (Figure 3H). This was also accompanied by an increase in its clustering. The treatment did not alter the number of clusters detected at the cell surface (Figure 3I). However, lowering [Cl^−^]_i_ increased by 1.12-fold the size of NKCC1 clusters (Figure 3J) and by 1.4-fold their fluorescence intensity (Figure 3K). These results indicate a chloride-dependent regulation of NKCC1 diffusion-capture, consistent with homeostatic regulation of the transporter.

### 3.4. The WNK Signaling Pathway Regulates NKCC1

The chloride-sensitive WNK signaling pathway regulates NKCC1 activity [21]. We assessed the role of this pathway in NKCC1 clustering using overexpression of constitutively active (WNK-CA) or kinase-dead (WNK-KD) WNK1 [24] and using WNK1 (WNK463) inhibitor. Overexpression of WNK-CA has no effect on the area explored by individual QDs (Figure 4A) or by hundreds of QDs (Figure 4B,C). Similarly, overexpression of WNK-CA did not alter the level of NKCC1 detected on the surface (Figure 4D,E) or the number of NKCC1 clusters (Figure 4F), or the size and intensity of these clusters (Figure 4G,H). Based on these results, we concluded that, under basal activity conditions, activation of the WNK1 pathway does not affect the diffusion, amount, and distribution of NKCC1 at the dendritic surface in mature hippocampal neurons.

Conversely, we studied the effects of inhibition of the WNK1 signaling pathway on NKCC1 surface expression and clustering following overexpression of WNK-KD or after acute exposure to a pan-WNK antagonist (WNK-463). An acute blockade for 30 min of WNK1 with WNK463 had no effect on the membrane stability of NKCC1 (Figure 4I,J), while blocking WNK1 activity for 7 DIV by overexpressing WNK-KD significantly reduced the membrane stability of NKCC1 (Figure 4I,J). On the other hand, WNK inhibition using genetic or pharmacological approaches significantly altered the clustering of NKCC1 by decreasing, respectively, to 2-fold and 1.6-fold the number of NKCC1 clusters (Figure 4K). This was not followed by a reduction in cluster size upon WNK463 treatment or WNK-KD overexpression (Figure 4L). However, WNK-KD overexpression decreased to 2-fold the intensity of NKCC1 clusters (Figure 4M). Therefore, inhibiting the WNK1 signaling pathway in basal activity conditions reduces NKCC1 membrane stability and clustering.

We then studied the contribution of the WNK1 effectors SPAK and OSR1 in the control of NKCC1 membrane diffusion, stability, and clustering using the SPAK/OSR1 inhibitor closantel [29]. Acute exposure of neurons to closantel rapidly reduced the dendritic exploration of individual QDs (Figure 5A). This was accompanied by a 1.14-fold reduction in NKCC1 diffusion coefficients (Figure 5B) and by a 1.28-fold decrease in its explored area (Figure 5C), revealing increased NKCC1 diffusion constraints as compared with control. This effect on diffusion was, however, not accompanied by a change in the surface detection of NKCC1 (Figure 5D,E), nor by a change in its clustering as determined by standard epifluorescence on the number of NKCC1 clusters (Figure 5F), as well as on their size and intensity (Figure 5G,H). However, the analysis of NKCC1 clusters using super-resolution imaging (Figure 5I) revealed that closantel exposure was reduced by 1.3-fold the cluster size (Figure 5J). This effect was not linked to a change in the number of clusters per dendritic length (Figure 5K). However, closantel increased by 1.75-fold the density of molecules per cluster (Figure 5L) as compared with untreated cells, indicative of molecular compaction. Therefore, the closantel-induced confinement of NKCC1 is accompanied by a rapid alteration in the nanoscale organization of the transporter. These results report that the WNK1/SPAK/OSR1 pathway regulates the membrane dynamics, stability, and clustering of NKCC1 in mature neurons. The fact that WNK1 activation (by overexpressing WNK-CA) has no effect on NKCC1 membrane dynamics, expression, and clustering suggests that this pathway is active in mature neurons and regulates NKCC1 diffusion-capture.

### 3.5. The WNK Signaling Pathway Targets Key Threonine Residues on NKCC1

WNK kinases promote the phosphorylation of NKCC1 T203/T207/T212 [24], which in turn results in NKCC1 activation [21]. To test the involvement of NKCC1-T203/207/212/217/230 phosphorylation status in the control of NKCC1 diffusion, we expressed NKCC1 mutations of T203/T207/T212 or T203/T207/T212/T217/T230 into alanine (TA3 and TA5, respectively) that mimic dephosphorylated states. The mobility and exploration of individual NKCC1 T203/207/212/217/230A were decreased relative to WT, especially for extrasynaptic QDs (Figure 6A). This was reflected by a 1.2-fold lower speed (Figure 6B), and a 1.48-fold increased confinement (Figure 6C) of QDs in the extrasynaptic membrane without changing the diffusion coefficient or the surface area explored at the inhibitory and excitatory synapses (Figure 6B,C). Therefore, the dephosphorylation of NKCC1 on key threonine residues confines the transporter in the extrasynaptic membrane. In agreement with a regulation of NKCC1 diffusion-capture by the WNK1 signaling pathway, these results indicate that a proportion of NKCC1 is phosphorylated on T203/207/212 in mature neurons. This differs from the KCC2 transporter, for which regulation by the WNK1 signaling was only observed when GABA_A_R activity was challenged [18].

We previously showed, in similar cultures, that an acute application of muscimol increases [Cl^−^]_i_ [18]. This is corroborated by the inhibition of WNK1 and dephosphorylation of NKCC1 [18]. Knowing that the phosphorylation of NKCC1 by WNK1 regulates its activity in non-neuronal cells [30], we wanted to know if the membrane stability and clustering of the mutated transporter were altered compared to that of the WT. Our results show that the surface pool of NKCC1 T203/T207/T212A is decreased (by 1.1-fold) compared to that of the WT transporter (Figure 6D,E). This decrease in the membrane stability of the transporter was accompanied by a 1.6-fold and a 1.7-fold decrease in the cluster density of NKCC1 T203/T207/T212A and NKCC1 T203/T207/T212/T217/T230A (Figure 6F), compared to the WT. The remaining NKCC1 T203/T207/T212A and NKCC1 T203/T207/T212/T217/T230A clusters were not changed in size or fluorescence intensity compared to WT (Figure 6G,H). This effect is reminiscent of that observed upon muscimol treatment (Figure 2) or WNK1 inhibition (Figure 4). Importantly, NKCC1 T203/T207/T212/T217/T230A prevented the muscimol-induced decrease in NKCC1 clustering (Figure 6I–K). We conclude that GABA_A_R-mediated regulation of NKCC1 membrane, diffusion, clustering, and stability involves phosphorylation of its T203/T207/T212/T217/T230 residues.

### 3.6. Functional Impact of NKCC1 Regulation by the WNK Pathway in Mature Hippocampal Neurons

Our work describes a regulation of membrane stability and clustering of NKCC1 by GABAergic activity. This regulation involves the phosphorylation of the transporter by the WNK/SPAK/OSR1 signaling pathway. To assess the functional relevance of this regulation, we looked at the impact of NKCC1 phosphorylation on [Cl^−^]_i_. We used SuperClomeleon [25] to quantify potential changes in intracellular chloride concentration. Changes in concentration were inferred from changes in YFP/CFP ratios (Figure 7A). This strategy has been shown to be effective in measuring variations in intracellular chloride levels in neurons. Indeed, we estimated in previous work the effect of gabazine and muscimol on [Cl^−^]_i_ showing a respective decrease and increase in chloride levels upon drug treatment [18]. As a control condition, we compared the YFP/CFP ratio of cells transfected with KCC2-WT vs. KCC2-T906/T1007E, i.e., a construction mimicking the phosphorylated state of the transporter with a reduced capacity to extrude chloride ions, notably by modifying its membrane stability and clustering [18,24]. An important decrease in the ratio was observed in cells transfected with KCC2-TE as compared to KCC2-WT, reflecting a higher [Cl^−^]_i_ (Figure 7B). Thus, this approach allows the measurement of [Cl^−^]_i_ elevation resulting from manipulations of chloride cotransporter membrane expression and, thereby, function. We then tested whether the expression of endogenous or recombinant NKCC1 significantly impacted [Cl^−^]_i_ in our neuronal preparation. First, we determined the impact of an acute blockade of NKCC1 with bumetanide (5 µM) on the YFP/CFP ratio in neurons transfected with SuperClomeleon alone. The ratio was comparable in both bumetanide-exposed and non-bumetanide-exposed neurons (Figure 7C). These results are in agreement with data suggesting that, at rest, endogenous NKCC1 does not significantly influence [Cl^−^]_i_ in mature hippocampal neurons [31,32]. Similarly, expression of the recombinant NKCC1-WT in mature neurons did not increase [Cl^−^]_i_ compared to neurons expressing the chloride probe alone, nor did it increase their sensitivity to bumetanide (Figure 7C). This suggests that the neuron tightly regulates the level of recombinant NKCC1 present at the cell membrane.

If the expression of NKCC1-WT does not influence [Cl^−^]_i_, then it is not surprising that a loss of function of the transporter cannot be detected. Indeed, we found that NKCC1-T203/T207/T212/T217/T230A, which shows a defect in membrane clustering compared to WT, has no effect on either the YFP/CFP ratio or the response to bumetanide (Figure 7C). However, since NKCC1 plays an important role on [Cl^−^]_i_ in mature neurons under conditions where KCC2 is down-regulated [4,9,18,32], we performed additional analyses on neurons co-transfected with the mutant transporter KCC2-T906/1007E, which has a reduced chloride extrusion capacity [24] and Figure 7A. Overexpression of KCC2-T906/1007E was preferred to the expression of an shRNA against KCC2 because this strategy led to neuronal death when NKCC1 was expressed in concert. Interestingly, neuronal death was not observed when the shRNA against KCC2 was expressed (not shown). This indicates that the recombinant NKCC1 transporter is functional in neurons and that the influx of chloride through NKCC1 in neurons in the absence of chloride ion extrusion capacity is toxic. In agreement with previous works [31,32], this result also means that in mature hippocampal neurons, KCC2 is the major regulator of [Cl^−^]_i_. However, no effect of the endogenous NKCC1 or of the recombinant NKCC1-WT or NKCC1-T203/T207/T212/T217/T230A was observed on the YFP/CFP ratio, nor on bumetanide sensitivity in conditions of low KCC2 activity (Figure 7D). We concluded that at rest, in conditions of normal or reduced KCC2 expression, endogenous or exogenous NKCC1 transporters are not significantly contributing to [Cl^−^]_i_ in mature hippocampal cultured neurons.

Since NKCC1 expression is increased in the adult epileptic brain and this expression contributes to increasing seizure susceptibility [3,7], we tested the contribution of the WNK1 signaling in the control of the membrane stability and function of NKCC1 in pathological conditions. For this purpose, neurons were acutely exposed to the convulsing agent 4-Amminopyridine (4-AP), a blocker of the voltage-dependent K^+^ channels responsible for membrane repolarization. This experiment was performed in the absence of TTX + KYN + MCPG. The “4-AP condition” was compared to the control condition in the absence of any drug. We previously showed that acute exposure of hippocampal neurons to 4-AP induces KCC2 endocytosis [13]. Conversely, we show here that this treatment rapidly increases the membrane expression of NKCC1 (Figure 7E). Pre-treatment of neurons with the inhibitor WNK463 prevented the 4-AP-induced increase in NKCC1 surface expression (Figure 7E), implicating WNK1 in the upregulation of NKCC1 at the neuronal surface.

Although we have shown that NKCC1 does not participate in the regulation of intracellular chloride levels under basal activity conditions in mature hippocampal neurons (Figure 7C,D), we show here the contribution of NKCC1 to neuronal chloride homeostasis under pathological conditions. Indeed, chloride imaging revealed that acute exposure to 4-AP induced a significant increase in intracellular chloride levels in neurons (Figure 7F,G). This effect was blocked by pre-incubating the neurons with the WNK antagonist or by blocking NKCC1 activity with bumetanide (Figure 7F,G), thus directly implicating the WNK signaling and NKCC1 in this regulation. Thus, we propose that 4-AP-induced hyperactivity activates the WNK pathway, which by phosphorylating NKCC1 on key threonine residues, increases its membrane expression and clustering, leading to intracellular chloride influx and decreased efficacy of GABAergic transmission.

## 4. Discussion

We studied, in mature hippocampal neurons, the cellular and molecular mechanisms regulating the cotransporter NKCC1, which transports chloride ions inside neurons. We showed that the transporter displays a heterogeneous distribution at the plasma membrane: it is either diffusely distributed and freely mobile in the membrane or organized in clusters where it is slowed down and confined. Here, we show that this distribution and behavior can be rapidly tuned by GABAergic activity changes. In particular, acute GABA_A_R activation or inhibition causes the escape of transporters from membrane clusters and their confinement in endocytic zones. GABA_A_R-mediated regulation of NKCC1 membrane distribution uses chloride as a secondary messenger and the Cl^−^-sensitive WNK1 pathway, which in turn affects the phosphorylation of key threonine residues on NKCC1. At rest, these modifications have little effect on [Cl^−^]_I_, but they participate in Cl^−^ accumulation in neurons in pathological conditions associated with an up-regulation of NKCC1 surface expression/function. 

An increase in clustering generally correlates with a slowing down and confinement of the molecule in a sub-cellular compartment, e.g., synapses for neurotransmitter receptors. Conversely, a dispersion of molecules from clusters implies a lifting of the diffusion brakes. Usually, transmembrane proteins are confined in membrane clusters due to their binding to scaffolding proteins that anchor them to the cytoskeleton. However, they can escape from these confined regions by lateral diffusion. This is the case of excitatory glutamate receptors and inhibitory GABA_A_Rs [33,34], as well as of ion transporters such as the chloride cotransporter KCC2 [11,12]. However, we found that under basal activity conditions, a proportion of NKCC1 transporters are slowed down and confined to endocytic zones [35]. The confinement of NKCC1 in endocytic zones is further increased following neuronal exposure to gabazine or muscimol without leading to its internalization. This suggests that under certain activity conditions, NKCC1 would be stored in the endocytic zones from which it could, however, come out and diffuse into the membrane to join membrane clusters.

Acute blockade of glutamatergic activity by the TTX + KYN + MCPG drug cocktail confines NKCC1 to the axon [35]. This suggests that spontaneous glutamatergic activity conversely makes NKCC1 mobile along the axon. Here we investigated the role of GABAergic transmission on NKCC1 diffusion in the axon. We show that blocking GABA_A_R activity with gabazine in the presence of TTX + KYN + MCPG to prevent the indirect effects of gabazine on increased excitation removes NKCC1 diffusion constraints in the axon. Conversely, activation of GABA_A_R by muscimol in the presence of TTX + KYN + MCPG slows NKCC1 in the axon. Thus, GABAergic and glutamatergic transmission have opposite effects on NKCC1 diffusion in the axon. We propose that glutamatergic activity regulates by a homeostatic mechanism the axonal diffusion of NKCC1 to compensate for the increase in activity by decreasing the depolarizing/excitatory action of GABA_A_R in the axon. Here, muscimol activation of GABA_A_Rs confines NKCC1 in the axon to enhance the depolarizing action of GABA, which would potentially impact neurotransmitter release [36] and action potential firing [31].

The effects of gabazine on NKCC1 diffusion in the dendrite and axon are opposite, highlighting distinct regulatory mechanisms. In the case of NKCC1 regulation by glutamatergic transmission, different effects were also observed in the dendrite and axon [35]. Future experiments will determine whether this difference is due to variations in intracellular chloride concentration (with higher concentration in the axon than in the dendrite) and activation of the WNK pathway or to other mechanisms. 

In the dendrites of mature neurons, we observed a similar effect of GABA_A_R activation or inhibition on the diffusion and clustering of dendritic NKCC1. In both cases, the transporter was sent to endocytic zones and confined there. The fact that there was no change in the global pool (surface + intracellular) of the transporter indicates that it is stored in the endocytic zones without being internalized and degraded. These endocytic zones have been shown to constitute reserve pools of neurotransmitter receptors, which can, depending on the synaptic demand, be released and reintegrated into the diffusing pool of receptors [37]. In the case of NKCC1, this reserve pool would allow a rapid increase in the transporter availability in the plasma membrane, for example, in pathological situations in which an up-regulation of NKCC1 membrane function has been observed [9].

By activating the GABA_A_R, muscimol raises [Cl^−^]_i_ [18]. An increase in [Cl^−^]_i_ blocks the activity of WNK1 and SPAK, OSR1 kinases [38,39], leading to the dephosphorylation of NKCC1-T203/207/212/217/230 [40,41,42] and reduced transporter activity [43,44]. We showed that following exposure to muscimol, NKCC1 escaped from membrane clusters and was confined in endocytic zones. Thus, the transition of the transporter between membrane clusters and endocytic zones by lateral diffusion would allow for modulating its availability in the membrane and its activity rapidly. The effects of muscimol are compatible with NKCC1-T203/207/212/217/230 dephosphorylation. Pharmacological (WNK-463 or closantel) or genetic (WNK-KD) blockade of the WNK pathway or the expression of NKCC1 TA3 or NKCC1 TA5 mutants that mimic NKCC1 dephosphorylation have the same effects as muscimol: they restrict NKCC1 in their movement and reduce the membrane clustering of the transporter. The demonstration that the effect of muscimol directly involves dephosphorylation of NKCC1-T203/207/212/217/230 was provided by the fact that the effect of muscimol on the clustering of NKCC1 can be prevented when the mutant NKCC1-TA5 was exposed to the drug, compared to WT.

In contrast, treatment of neurons with gabazine, by blocking the activity of GABA_A_Rs, decreases dendritic [Cl^−^]_i_. In non-neuronal cells, low chloride activates WNK1/SPAK [38,39] by auto-phosphorylation of WNK1 S382 residue. Then, WNK phosphorylates in cascade SPAK on S373 and OSR1 on S325 [45] that in turn phosphorylate NKCC1 on T203/207/212/217/230 and increase the surface expression and activity of the transporter [44,46]. We showed that this signaling cascade is operant in mature hippocampal neurons: gabazine activates the WNK1/SPAK/OSR1 pathway by phosphorylation, thus inducing the phosphorylation of KCC2-T906/1007 as well as NKCC1-T203/207/212/217/230 [18]. If muscimol decreases NKCC1 clustering and confines it to endocytic zones, gabazine treatment should conversely induce the escape of the transporter from endocytic zones, thereby increasing its clustering and function in the membrane. Although we observed that a decrease in [Cl^−^]_i_ by substituting chloride with methane sulfonate decreases the confinement of NKCC1 and increases its membrane stability and clustering, the treatment of neurons with gabazine did not reproduce this effect. On the contrary, gabazine confined the transporter and induced the loss of its clustering just as muscimol did. However, we showed that antagonizing inhibition with gabazine reduces the surface expression of KCC2 by increasing the lateral diffusion and internalization of the transporter [18]. This results in reduced intracellular chloride extrusion capacity of the neuron leading to a significant increase in [Cl^−^]_i_ as monitored by SuperClomeleon imaging [18]. KCC2 is more effective in regulating [Cl-]_i_ than NKCC1 in dendrites of mature neurons [47,48]. We, therefore, hypothesize that the regulation of NKCC1 diffusion-capture by gabazine is not due to a reduction in [Cl^−^]_i_ but instead to an increase in [Cl^−^]_i_ following KCC2 regulation by the WNK1/SPAK/OSR1 pathway. Thus, changes in the membrane expression of KCC2 (under the control of the WNK1 pathway) would condition that of NKCC1, thus allowing [Cl^−^]_i_ to be maintained at a low level in mature neurons. This would explain why expressing recombinant NKCC1-WT in mature neurons does not significantly increase its expression at the membrane, nor does it increase [Cl^−^]_i_. This suggests that the membrane expression level of NKCC1 is under the control of KCC2 and its tuning of [Cl^−^]_i_.

Nevertheless, we showed that the membrane stability and clustering of NKCC1 can be rapidly regulated by lateral diffusion and that this mechanism is rapidly controlled by GABAergic inhibition and the WN1K/SPAK/OSR1 pathway on the dendrites of mature neurons. Although this pathway has little influence on the amount/function of NKCC1 at the neuronal surface under basal activity conditions, we propose that it may play a role in pathological situations associated with increased expression levels of NKCC1 based on our 4-AP data. Interestingly, in the pathology, the upregulation of NKCC1 is often accompanied by a down-regulation of KCC2 at the neuronal surface [3,16]. KCC2 is also regulated by diffusion-capture. We showed that a short exposure of neurons to the convulsive agent 4-AP increases the lateral diffusion of KCC2, which escapes from the clusters, is internalized and degraded [13]. Thus, lateral diffusion would be a general mechanism to control the membrane stability of chloride cotransporters. Moreover, the fact that KCC2 is also regulated in mature neurons by the WNK1 pathway [18] and that this regulation has an inverse effect on membrane stability, clustering, and function of KCC2 indicates that this pathway is a target of interest in the pathology. Inhibition of the pathway would prevent the loss of KCC2 and the increase in NKCC1 at the surface of the neuron, thus preventing the abnormal rise of [Cl^−^]_i_ in the pathology and the resulting adverse effects.

## Figures and Tables

**Figure 1 cells-12-00464-f001:**
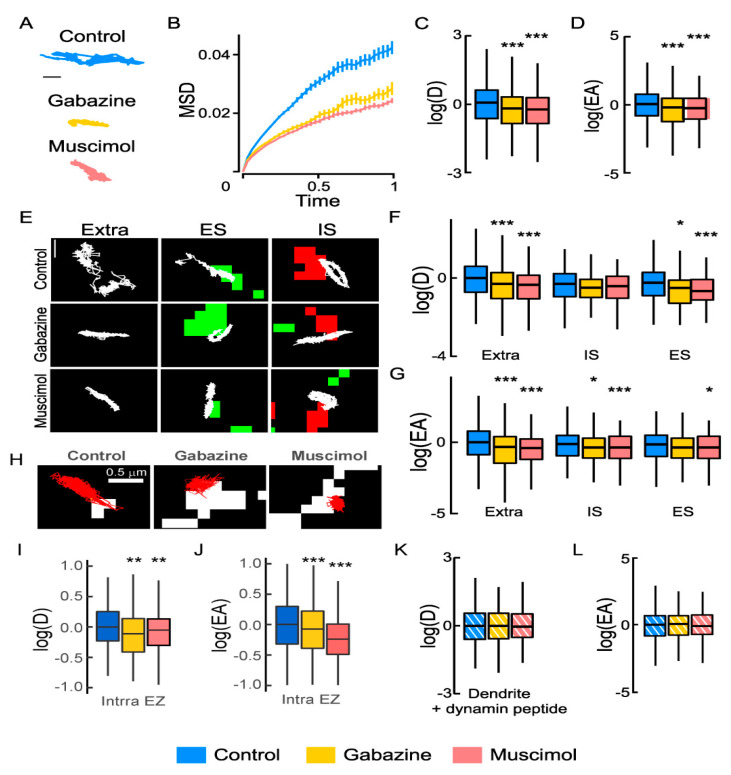
GABA type A receptor (GABA_A_R) activity regulates the Na^+^-K^+^-2Cl^−^ cotransporter NKCC1 membrane dynamics: (**A**) Individual NKCC1 trajectories with reduced surface exploration upon gabazine or muscimol treatment. Scale bar, 0.5 µm. (**B**) Time-averaged mean square displacement (MSD) functions in control (blue) vs. gabazine (yellow) or muscimol (orange) conditions show increased confinement upon gabazine or muscimol application. (**C**,**D**) Boxplots of log diffusion coefficient (D) of NKCC1 in control condition (blue) or upon application of gabazine (yellow) or muscimol (orange) showing reduced diffusion upon drug treatments. N = 1558 quantum dots (QDs) (control, 41 cells), *n* = 387 QDs (gabazine, 18 cells), Welch *t*-test, *p* = 1.3 × 10^−6^, *n* = 545 QDs (muscimol, 27 cells), Welch *t*-test, *p* = 2.1 × 10^−12^, 5 cultures. (**D**) Median explored area EA in control vs. gabazine or muscimol conditions show increased confinement upon gabazine (Welch *t*-test, *p* = 2 × 10^−8^) or muscimol (Welch *t*-test, *p* = 4.2 × 10^−11^) treatment. (**E**) Trajectories (white) overlaid with clusters of homer1c-DsRed (green) or gephyrin-Finger-YFP (red) to identify extrasynaptic trajectories (extra), trajectories at excitatory (ES), and inhibitory synapses (IS). Scale bar, 0.4 µm. (**F**,**G**) Log(D) (**F**) and EA (**G**) of NKCC1 are decreased upon gabazine or muscimol application as compared with control condition. Note that the effect is more pronounced for extrasynaptic trajectories than for ES or IS trajectories. Diffusion coefficient (D): Extra, *n* = 899 QDs (control), 227 QDs (gabazine), *p* = 2.9 × 10^−5^, *n* = 268 QDs (muscimol) *p* = 2.2 × 10^−9^; IS, *n* = 244 QDs (control), 79 QDs (gabazine) *p* = 0.16; *n* = 142 QDs (muscimol) *p* = 0.19; ES, *n* = 415 QDs (control), *n* = 81 QDs (gabazine) *p* = 0.021, *n* = 135 QDs (muscimol) *p* = 0.00046. Explored area (EA): Extra, gabazine *p* = 2.9 × 10^−5^, muscimol *p* = 2.2 × 10^−9^; IS, gabazine *p* = 0.16, muscimol *p* = 0.19; ES, *n* = 415 QDs (control), *n* = 81 QDs (gabazine) *p* = 2.1 × 10^−2^, *n* = 135 QDs (muscimol) *p* = *p* = 4.6 × 10^−4^. (**H**) NKCC1 trajectories (red) in control vs. gabazine or muscimol conditions in relation to endocytic zones identified by clathrin-YFP clusters (white). Scale bar, 0.5 µm. (**I**,**J**) Reduced diffusion coefficient (**I**) and explored area (**J**) of NKCC1 within endocytic zones (Intra EZ) upon muscimol and gabazine treatment. Diffusion coefficient (D): Ctrl *n* = 117 QDs, 48 cells, Gbz *n* = 145 QDs, 64 cells, *p* = 1.4 × 10^−3^; Ctrl *n* = 273 QDs, 35 cells, Musc *n* = 247 QDs, 28 cells, *p* = 5.6 × 10^−3^, 3 cultures. Explored area (EA): Gbz, *p* = 5.46 × 10^−5^ and Musc, *p* = 2.2 × 10^−16^, 3 cultures. (**K**,**L**) No effect of gabazine or muscimol on log(D) (**K**) and EA (**L**) of dendritic NKCC1 in conditions of blockade of endocytosis (+ dynamin peptide). Diffusion coefficient (D): Bulk Ctrl *n* = 491 QDs, Gbz *n* = 238 QDs, *p* = 0.6 and Musc *n* = 243 QDs, *p* = *p* = 6.1 × 10^−1^, 5 cultures. Explored area (EA): Gbz, *p* = 0.84 and Musc, *p* = 8.1 × 10^−1^, 5 cultures. (**B**,**F**,**I**,**K**) D in µm^2^.s^−1^; C: MSD in µm^2^ vs. time(s); (**D**,**G**,**J**,**L**) EA in µm^2^. In all graphs, *, *p* < 5.0 × 10^−2^; **, *p* < 1.0 × 10^−2^; ***, *p* < 1.0 × 10^−3^.

**Figure 2 cells-12-00464-f002:**
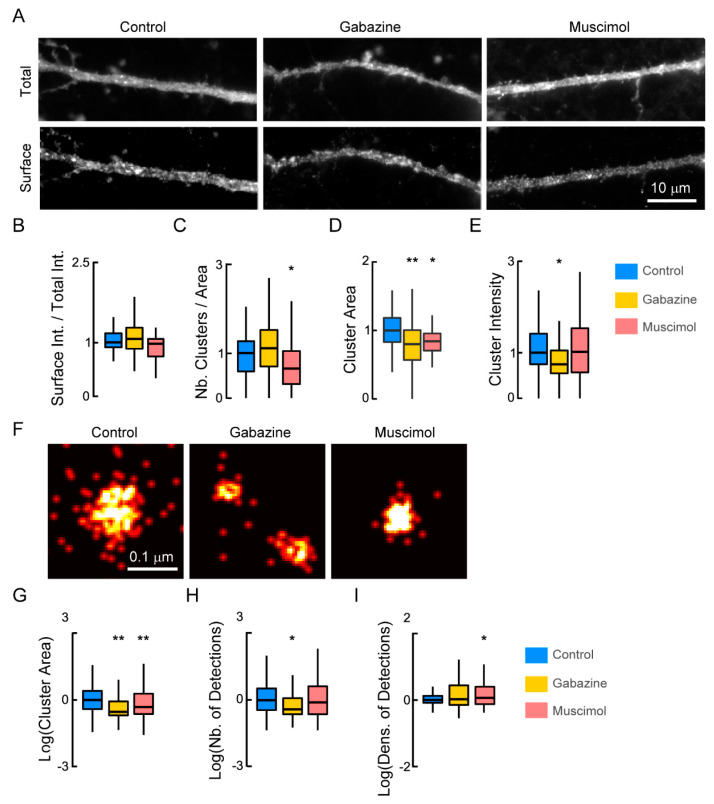
Regulation of NKCC1 membrane clustering by GABA_A_R-mediated inhibition: (**A**) Conventional microscopy showing NKCC1 surface staining in hippocampal neurons at 21 days in vitro (DIV) in absence (Control) or presence of gabazine (Gabazine) or muscimol (Muscimol) for 30 min. Scale bar, 10 µm. (**B**) Quantification of the ratio of the surface/total pool of NKCC1 in control (blue), gabazine (yellow), and muscimol (orange) conditions showing no changes in the surface level of NKCC1 after gabazine or muscimol treatment. Ctrl *n* = 30 cells, Gbz, *n* = 41 cells, MW test *p* = 8.7 × 10^−1^, Musc *n* = 38 cells, *p* = 1.7 × 10^−1^, 8 cultures. (**C**–**E**), Quantification of NKCC1 cluster number (**C**), area (**D**), and intensity (**E**) showing reduced density and size of NKCC1 clusters upon muscimol treatment, while gabazine treatment reduced the size and intensity of NKCC1 clusters. Values were normalized to control values. MW test: Gabazine: cluster number (nb) *p* = 4.9 × 10^−1^, area *p* = 3.0 × 10^−3^, 0.003, intensity *p* = 1.7 × 10^−2^. Muscimol: cluster nb *p* = 2.7 × 10^−2^, area *p* = 2.9 × 10^−2^, intensity *p* = *p* = 9.2 × 10^−1^. (**F**–**I**) Stochastic Optical Reconstruction Microscopy (STORM) showing that gabazine and muscimol treatments alter NKCC1 nanoclusters at the surface of hippocampal neurons. (**F**) Representative STORM images of NKCC1 at the surface of neurons exposed 30 min to gabazine or muscimol. Scale bar, 0.1 µm. (**G**) Quantification of NKCC1 cluster area shows reduction in nanocluster size upon gabazine and muscimol treatment. Ctrl *n* = 550 nanoclusters, Gbz *n* = 192 nanoclusters, Monte Carlo simulations of the MW test *p* = 2.0 × 10^−3^, Musc *n* = 410 nanoclusters, *p* = 0.004, 4 cultures. (**H**) Quantification of the number of particles detected per nanocluster showing reduced number of detection upon gabazine (Monte Carlo simulations of MW test, *p* = 1.2 × 10^−2^) but not muscimol (*p* = 3.0 × 10^−3^) exposure. (**I**) Quantification of the density of NKCC1 molecules per square nanometer highlighting denser NKCC1 packing upon neuronal exposure to muscimol (Monte Carlo simulations of MW test, *p* = 1.6 × 10^−2^) but not gabazine (*p* = 9.7 × 10^−1^). (**C**) µm^−1^, (**D**) µm², (**G**) nm², (**H**) µm^−1^, (**I**) µm^−2^. In all graphs, *, *p* < 5.0 × 10^−2^; **, *p* < 1.0 × 10^−2^.

**Figure 3 cells-12-00464-f003:**
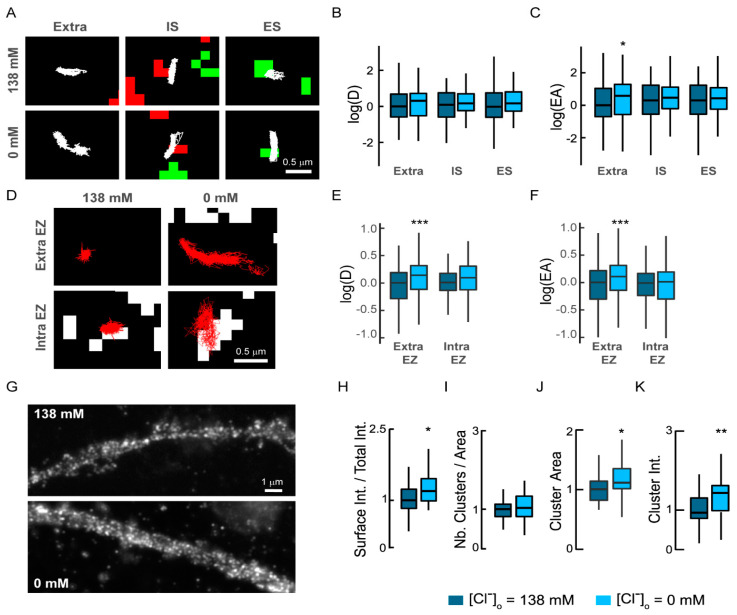
Lowering intracellular chloride increases the membrane diffusion, clustering, and stability of NKCC1: (**A**) Trajectories of NKCC1 (white) under low and high Cl^−^ concentration in the extrasynaptic area (extra) and at inhibitory (IS) or excitatory (ES) synapses. Scale bar, 0.5 µm. (**B**) No major effect of a reduction in Cl^−^ concentration on log(D) (**B**) of NKCC1. Note the increase in log(EA) for extrasynaptic trajectories (**C**) in conditions of low chloride. Diffusion coefficient (D): extra, low Cl^−^ *n* = 128 QDs, high Cl^−^ *n* = 93 QDs, Welch *t*-test *p* = 8.9 × 10^−1^; IS, low Cl^−^ *n* = 64 QDs, high Cl^−^ *n* = 56 QDs, Welch *t*-test *p* = 7.4 × 10^−1^; ES, low Cl^−^ *n* = 62 QDs, high Cl^−^ *n* = 68 QDs, Welch *t*-test *p* = 3.6 × 10^−1^, 2 cultures. Explored area (EA): extra, low Cl^−^ Welch *t*-test *p* = 1.0 × 10^−2^; IS, Welch *t*-test *p* = 6.0 × 10^−1^; ES, Welch *t*-test *p* = 0.6. (**D**) Examples of NKCC1 trajectories (red) inside and outside clathrin-YFP fluorescent (white) endocytic zones in conditions of low (0 mM) vs. high (138 mM) Cl^−^ concentration. Scale bar, 0.5 µm. (**E**,**F**) lowering intracellular chloride levels increases log(D) and log(EA) of NKCC1 located outside endocytic zones (Extra EZ) while the diffusion of NKCC1 inside endocytic zones (Intra EZ) is unchanged. Diffusion coefficient (D): extra EZ, low Cl^−^ *n* = 209 QDs, high Cl^−^ *n* = 238 QDs, Welch *t*-test *p* = 6.0 × 10^−4^; intra EZ, low Cl^−^ *n* = 91 QDs, high Cl^−^ *n* = 105 QDs, Welch *t*-test *p* = 2.3 × 10^−1^, 2 cultures. Explored area (EA): extra EZ, *p* = 7.12 × 10^−8^; intra EZ, *p* = 4.1 × 10^−1^. (**G**–**K**) Lowering intracellular chloride increases surface detection and clustering of NKCC1. **G**, HA surface staining in neurons expressing recombinant NKCC1-HA in conditions of high vs. low Cl^−^ concentration for 30 min. Scale bar, 1 µm. (**H**) Quantification of the surface/total pool of NKCC1 in high (dark blue) and low (light blue) Cl^−^ concentration showing increase in NKCC1 surface staining upon reduced Cl^−^ level. Low Cl^−^ *n* = 35 cells, high Cl^−^ *n* = 40 cells, Welch *t*-test *p* = 1.0 × 10^−1^, 3 cultures. (**I**–**K**) Quantification of NKCC1-HA cluster number (**I**), area (**J**), and intensity (**K**) shows increased size and intensity of NKCC1 clusters upon reduction in Cl^−^ levels. Values normalized to control values. Cluster number (Nb) MW test *p* = 2.0 × 10^−1^, area MW test *p* = 1.8 × 10^−2^, intensity MW test *p* = 5.6 × 10^−3^. (**B**,**E**) D in µm^2^.s^−1^; (**C**,**F**) EA in µm^2^; (**I**) µm^−1^; (**J**) µm². In all graphs, *, *p* < 5.0 × 10^−2^; **, *p* < 1.0 × 10^−2^; ***, *p* < 1.0 × 10^−3^.

**Figure 4 cells-12-00464-f004:**
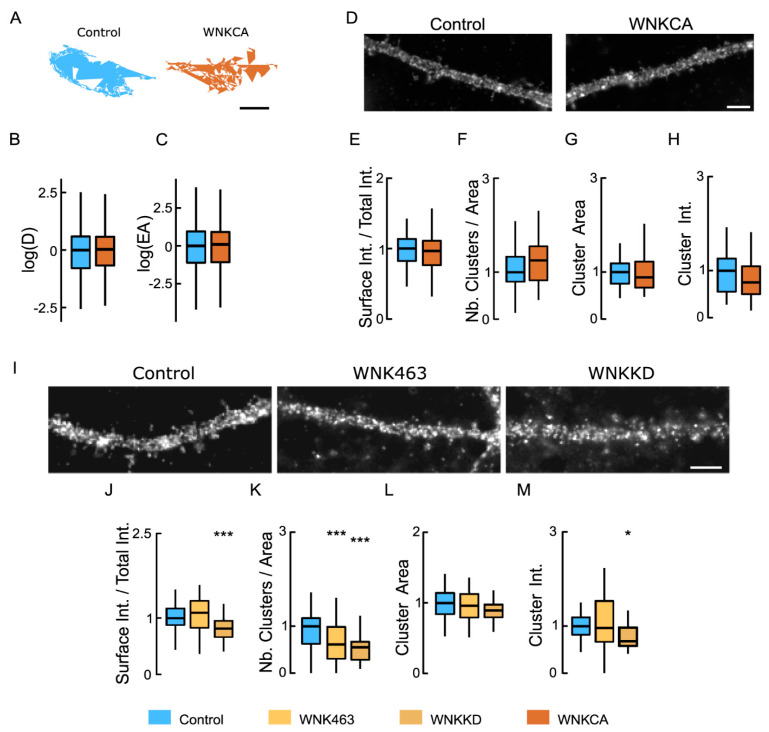
Inhibiting WNK1 reduces the clustering of NKCC1 at the neuronal surface: (**A**) Representative trajectories of NKCC1 in neurons expressing constitutively active WNK1 (WNK-CA, brown) vs. control (blue). Bar: 0.4 µm. (**B**,**C**) WNK-CA over-expression does not change NKCC1 diffusion at the plasma membrane, with no effect on D (**B**) and EA (**C**). Diffusion coefficient (D): Bulk, Ctrl *n* = 305 QDs, WNK-CA *n* = 358 QDs, Welch *t*-test *p* = 8.1 × 10^−1^, 3 cultures; Explored area (EA): Welch *t*-test *p* = 4.9 × 10^−1^. (**D**) HA surface staining in hippocampal neurons expressing recombinant NKCC1-HA together with “constitutively active” WNK1 (WNK1-CA) or a control plasmid. Scale bar, 4 µm. (**E**) Overexpressing WNK-CA does not modify the level of NKCC1 expressed at the cell surface. Ctrl *n* = 45 cells, WNK-CA: *n* = 43 cells, Welch *t*-test *p* = 2.6 × 10^−1^, 3 cultures. (**F**–**H**) No effect of WNK-CA expression on NKCC1 cluster number (**F**), area (**G**), and intensity (**H**). All values were normalized to the control values. Cluster number (Nb) MW test *p* = 1.4 × 10^−1^, area MW test *p* = 3.2 × 10^−1^, intensity MW test *p* = 0.25. (**I**) Impact of inhibition of WNK1 on HA surface staining of neurons transfected with NKCC1-HA. WNK1 was inhibited either by incubating the neurons for 30 min with a specific inhibitor, WNK-463, or by co-expressing kinase-dead WNK1 (WNK1-KD). Scale bar, 4 µm. (**J**) Incubating the neurons with WNK-463 (orange) does not modify the surface level of NKCC1 compared to controls (blue); however, over-expression of WNK-KD (brown) sharply reduced NKCC1 surface levels. WNK-463 experiment: Ctrl *n* = 24 cells, WNK-463 *n* = 25 cells, Welch *t*-test *p* = 5.8 × 10^−1^. WNK-KD experiment: Ctrl *n* = 34 cells, WNK-KD *n* = 24 cells, Welch *t*-test *p* = 9.6 × 10^−3^, *p* = 0.0096, 4 cultures. (**K**–**M**) Loss of NKCC1 clusters (**K**) and reduced cluster size (**L**) but not cluster intensity (**M**) upon WNK1 activity suppression with either WNK-KD or WNK-463, as compared to control conditions. Values were normalized to the corresponding control values. MW test: WNK-KD experiment: Cluster number (Nb) *p* = 4.5 × 10^−7^, area *p* = 1.2 × 10^−2^, intensity *p* = 3.1 × 10^−3^. WNK-463 experiment: Cluster number (Nb) *p* = 8.0 × 10^−4^, area *p* = 8.6 × 10^−1^, intensity *p* = 8.3 × 10^−1^. In all graphs, *, *p* < 5.0 × 10^−2^; ***, *p* < 1.0 × 10^−3^.

**Figure 5 cells-12-00464-f005:**
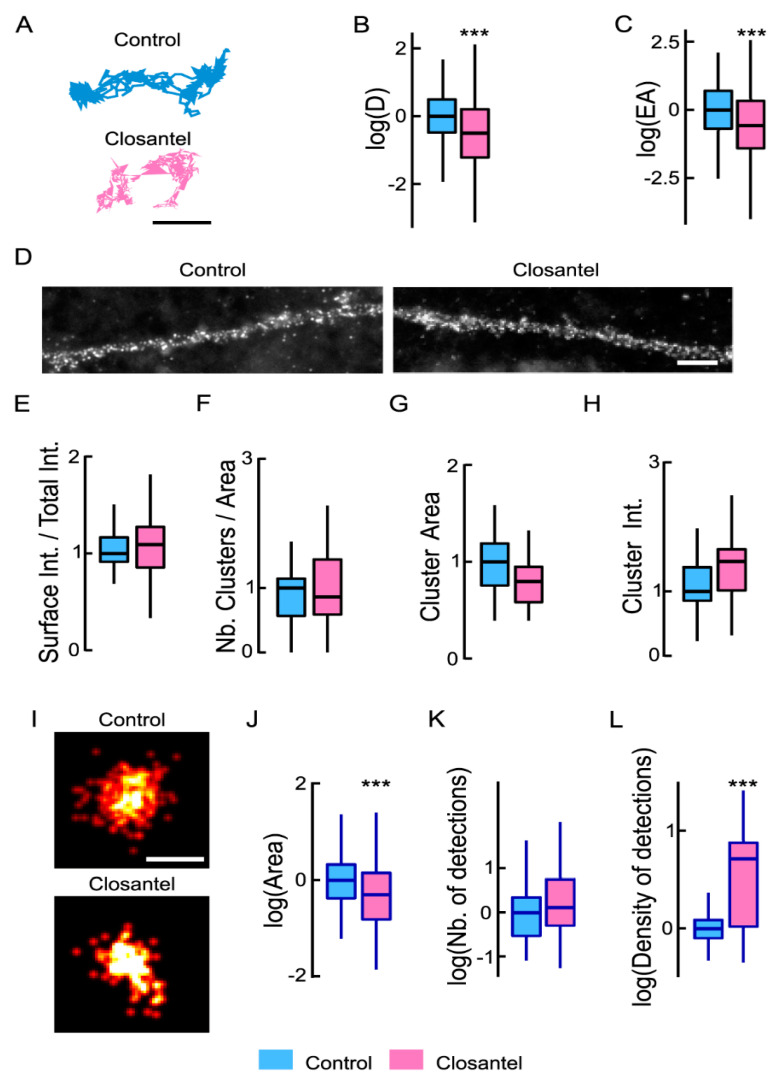
Inhibiting SPAK-OSR1 activity tunes NKCC1 membrane diffusion and clustering: (**A**) NKCC1 trajectories showing decreased exploration in the presence of closantel (pink) compared to connrol (blue). Scale bar, 0.5 µm. (**B**,**C**), Log (D) (**B**) and EA (**C**) of NKCC1 are decreased upon closantel application (pink) as compared with control condition (blue), indicating reduced mobility and increased confinement. Diffusion coefficient (D): Bulk, Ctrl *n* = 433 QDs, closantel *n* = 371 QDs, Welch *t*-test *p* = 5.5 × 10^−5^, 2 cultures. Explored area (EA): Welch *t*-test *p* = 4.8 × 10^−11^. (**D**–**H**), Standard epifluorescence microscopy showing no effect of closantel on NKCC1 membrane immunoreactivity. (**D**) HA surface staining in neurons expressing NKCC1-HA and treated or not with closantel. Scale bar, 4 µm. (**E**) Closantel treatment does not modify the surface level of NKCC1. Ctrl *n* = 51 cells, closantel *n* = 61 cells, Welch *t*-test *p* = 4.9 × 10^−1^, 7 cultures. (**F**–**H**) Closantel has no impact on NKCC1 cluster number (**F**), area (**G**), and intensity (**H**). Cluster Number (Nb) MW test *p* = 6.0 × 10^−1^, area MW test *p* = 9.0 × 10^−2^, intensity MW test *p* = 8.1 × 10^−1^. (**I**–**L**) STORM showing that closantel affects NKCC1 organization at the nanoscale. (**I**) Representative images of NKCC1 at the surface of neurons exposed 30 min to closantel vs. control condition. Scale bar, 0.1 µm. Quantification of NKCC1 cluster area (**J**), number of cluster (**K**) and density of detections in the cluster (**L**) reveal reduction in cluster size upon closantel treatment. Ctrl *n* = 218 clusters, closantel *n* = 147 clusters, 2 cultures. Cluster area: Monte Carlo simulations of the MW test *p* < 0.001; Nb detection: MW test *p* = 5.8 × 10^−2^, density: MW test *p* < 0.001. In all graphs, ***, *p* < 1.0 × 10^−3^.

**Figure 6 cells-12-00464-f006:**
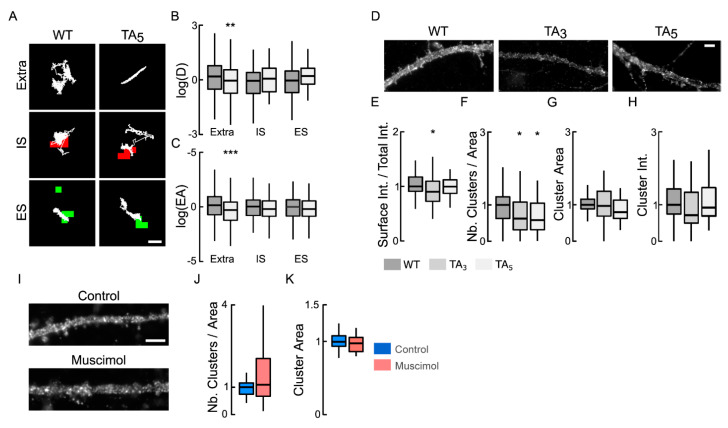
NKCC1 membrane dynamics, stability, and clustering are dependent on NKCC1 phosphorylation of T203/207/212/217/230. (**A**) Examples of NKCC1-T203/207/212/217/230 (WT) and NKCC1-T203/207/212/217/230A (TA_5_) trajectories (white) in resting condition at extrasynaptic site (extra), at inhibitory (IS) and excitatory glutamatergic (ES) synapses. Scale bar, 0.5 µm. (**B**,**C**) Log(D) (**B**) and EA (**C**) show that the dephospho-mimetic NKCC1-TA_5_ (light gray) is slower and more confined than NKCC1-WT (gray) at extrasynaptic sites but not near synapses. Diffusion coefficient (D): WT: extra, *n* = 189 QDs, IS, *n* = 42 QDs, ES *n* = 30 QDs; TA_5_: extra, *n* = 166 QDs, IS, *n* = 33 QDs, ES *n* = 34 QDs; from 67 cells and 2 cultures. Welch *t*-test: extra *p* = 8.8 × 10^−3^, IS *p* = 2.8 × 10^−1^, ES *p* = 7.3 × 10^−1^. Explored area (EA): extra *p* = 4.5 × 10^−4^, IS *p* = 7.1 × 10^−1^, ES *p* = 4.5 × 10^−1^. (**D**) HA surface staining in hippocampal neurons expressing recombinant NKCC1-WT (WT) vs. NKCC1-TA_3_ (TA3) or NKCC1-TA_5_ (TA5) in resting conditions. Scale bar, 4 µm. (**E**) Expression of NKCC1-TA3 is slightly reduced at the plasma membrane as compared to NKCC1-WT. WT (dark gray) *n* = 68 cells, TA_3_ (gray) *n* = 43 cells, TA_5_ (light gray) *n* = 36 cells (9 cultures); WT vs. TA_3_ Welch *t*-test *p* = 4.1 × 10^−2^, WT vs. TA_5_ *p* = 8.7 × 10^−1^, 9 cultures. (**F**–**H**) Quantification of cluster number (**F**), area (**G**), and intensity (**H**) for NKCC1-TA_3_, NKCC1-TA_5_ vs. NKCC1-WT. Note the reduced density of cluster (**F**) for NKCC1-TA_3_ and NKCC1-TA_5_ as compared to NKCC1-WT. Cluster number (Nb) WT vs. TA3 MW test *p* = 4.2 × 10^−2^, WT vs. TA_5_ MW test *p* = 1.4 × 10^−2^; area: WT vs. TA_3_ MW test *p* = 4.6 × 10^−1^, WT vs. TA_5_ MW test *p* = 2.2 × 10^−1^; intensity WT vs. TA_3_ MW test *p* = 3.0 × 10^−1^, WT vs. TA_5_ MW test *p* = 8.6 × 10^−1^. (**I**) HA surface staining in hippocampal neurons expressing recombinant NKCC1-TA_5_ exposed or not to muscimol. Scale bar, 4 µm. (**J**–**K**) Muscimol application has no effect on NKCC1-TA_5_ cluster number: MW test, *p* = 4.5 × 10^−1^, (**J**) and area: MW test, *p* = 3.8 × 10^−1^, (**K**) Control: 26 cells, muscimol: 22 cells, 3 cultures. In all graphs, *, *p* < 5.0 × 10^−2^; **, *p* < 1.0 × 10^−2^; ***, *p* < 1.0 × 10^−3^.

**Figure 7 cells-12-00464-f007:**
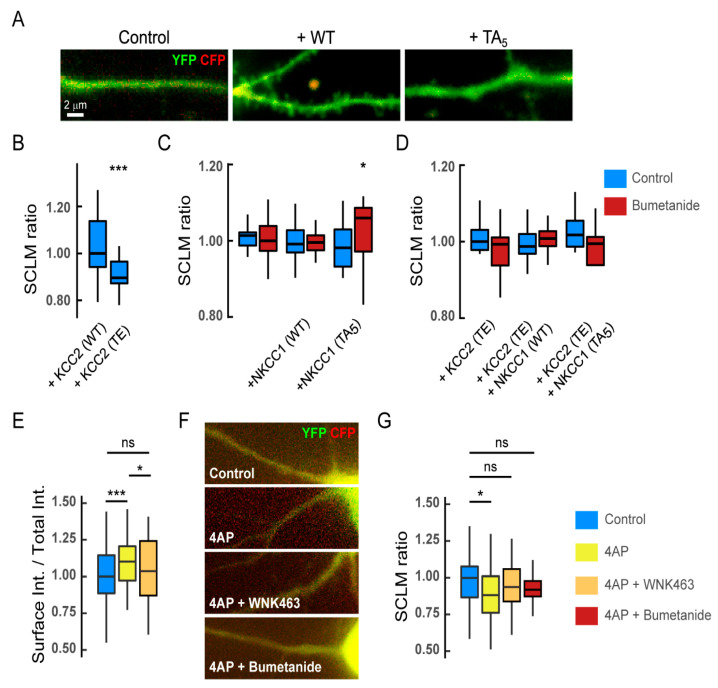
Functional impact of NKCC1 regulation by the WNK signaling on chloride homeostasis. (**A**–**D**) In basal activity conditions, expression of recombinant WT or dephospho-mimetic NKCC1 has no major impact on [Cl^−^]_i_ in mature hippocampal neurons: (**A**) composite images of CFP (red) and YFP (green) in neurons expressing SuperClomeleon (SCLM) alone (control) or in combination with NKCC1-T203/207/212/217/230 (WT) or NKCC1-T203/207/212/217/230A (TA_5_) in resting conditions. Scale bar, 2 µm. (**B**) CFP/YFP fluorescence ratio in hippocampal neurons expressing SCLM together with KCC2-T906/T1007 (WT) or phospho-mimetic KCC2-T906/T1007E (TE). KCC2-WT *n* = 36 cells, KCC2-TE *n* = 39 cells, MW test *p* = 7.1 × 10^−6^, 3 cultures. (**C**) CFP/YFP fluorescence ratio in neurons expressing SCLM alone or in combination with NKCC1-WT or NKCC1-TA_5_ before (blue) and after (red) 10–30 min application of bumetanide. SCLM Ctrl *n* = 20 cells, Bumet *n* = 17 cells; SCLM + NKCC1-WT Ctrl *n* = 18 cells, Bumet *n* = 20 cells; SCLM + NKCC1-TA_5_ Ctrl *n* = 21 cells, Bumet *n* = 16 cells; 3 cultures. Ctrl vs. Bumet: SCLM MW test *p* = 9.6 × 10^−1^, SCLM + NKCC1-WT MW test *p* = 5.3 × 10^−1^, SCLM + NKCC1-TA_5_ MW test *p* = 2.9 × 10^−2^. SCLM vs. SCLM + NKCC1-WT, MW test *p* = 7.1 × 10^−1^, SCLM vs. SCLM + NKCC1-TA_5_, MW test *p* = 6.9 × 10^−1^; SCLM + NKCC1-WT vs. SCLM + NKCC1-TA5, MW test *p* = 8.9 × 10^−1^. (**D**) CFP/YFP fluorescence ratio in neurons expressing the same recombinant proteins as in B but in the presence of KCC2-TE before (blue) and after (red) bumetanide treatment. SCLM + KCC2-TE Ctrl *n* = 15 cells, Bumet *n* = 15 cells, MW test *p* = 1.5 × 10^−1^; SCLM + KCC2-TE + NKCC1-WT Ctrl *n* = 13 cells, Bumet *n* = 16 cells, MW test *p* = 5.6 × 10^−1^; SCLM + KCC2-TE + NKCC1-TA_5_ Ctrl *n* = 21 cells, Bumet *n* = 8 cells, MW test *p* = 9.2 × 10^−2^; SCLM + KCC2-TE vs. SCLM + KCC2-TE + NKCC1-WT, MW test *p* = 6.5 × 10^−1^, SCLM vs. SCLM + KCC2-TE + NKCC1-TA_5_, MW test *p* = 2.0 × 10^−1^, SCLM + KCC2-TE + NKCC1-WT vs. SuperClomeleon + KCC2-TE + NKCC1-TA_5_, MW test *p* = 1.2 × 10^−1^. Two cultures. (**E**) Quantification of the surface/total pool of NKCC1 in control (blue), 4-AP (yellow), and 4-AP + WNK463 (orange) conditions showing a significant increase in surface NKCC1 upon 4AP treatment, an effect prevented by pre-incubation of neurons with WNK463 (orange). Ctrl *n* = 63 cells, 4-AP, *n* = 64 cells, MW test *p* = 4.6 × 10^−4^, 4-AP + WNK463 *n* = 57 cells, 4-AP vs. 4-AP + WNK463 *p* = 3.0 × 10^−2^, Ctrl vs. 4-AP + WNK463 *p* = 1.9 × 10^−1^, 4 cultures. (**F**) Overlay images of CFP (red) and YFP (green) in neurons expressing SuperClomeleon (SCLM) and NKCC1 in control vs. 4AP, 4AP + WNK463 or 4AP + bumetanide conditions. Scale bar, 2 µm. (**G**) CFP/YFP fluorescence ratio in neurons expressing SCLM and NKCC1 in control (blue) vs. 4-AP (yellow), 4-AP + WNK463 (orange), or 4-AP + bumetanide (red) conditions. Ctrl *n* = 53 cells, 4-AP *n* = 48 cells, MW test *p* = 1.0 × 10^−2^; 4-AP + WNK463 *n* = 41 cells, MW test *p* = 2.0 × 10^−1^; 4-AP + bumetanide *n* = 29 cells, MW test *p* = 2.1 × 10^−1^, 3–6 cultures. In all graphs, n, not significant; *, *p* < 5.0 × 10^−2^; ***, *p* < 1.0 × 10^−3^.

## Data Availability

The data that support the findings of this study are available from the corresponding author upon reasonable request. The transfer of plasmids generated for this study will be made available upon request. A Materials Transfer Agreement may be required.

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
