# Peer review of "Lateral Diffusion of NKCC1 Contributes to Chloride Homeostasis in Neurons and Is Rapidly Regulated by the WNK Signaling Pathway"

_cells, 2023, doi:10.3390/cells12030464_

Round 1
Reviewer 1 Report
The Manuscript “Lateral Diffusion of NKCC1 Contributes to Neuronal Chloride Homeostasis and is Rapidly Regulated by the WNK Signaling Pathway” by Come et al., analyzes the role of GABAergic inhibition in the regulation of the chloride transporter NKCC1 in cultured hippocampal neurons. In particular, the Authors employed different approaches including pharmacology, mutagenesis, immunocytochemistry, SPT, chloride imaging and STORM to demonstrate that the intracellular concentration of chloride ions and the WNK/SPAK pathway are involved in the chloride homeostasis through the modulation of NKCC1. In addition, they propose that such signaling pathways could be involved in the pathological increase of intracellular chloride observed in different forms of epilepsy.
The study is well conducted and tackle an important point concerning the cellular and molecular mechanisms responsible for the chloride homeostasis, and, therefore, the strength and polarity of GABAergic signaling. However, some points need clarification.
Major points:
11) Across the whole manuscript, it is not clear what “modulation of NKCC1” means: is that change in expression or function (or both)? Please clarify
22) The Authors report that the blocking of clathrin-mediated endocytosis prevented the effects of muscimol and gabazine on receptor mobility and clustering. Yet, they show that muscimol and gabazine do not alter receptor internalization. If endocytosis is not involved, why the block of endocytosis should alter the effect of muscimol and gabazine on NKCC1 diffusion? Does dynamin peptide only prevents the endocytosis of NKCC1 or the NKCC1 “docking” to endocytic zones? The block of endocytosis by dynamin per se should not straightforwardly explain the lack of muscimol and gabazine effects on NKCC1 dynamics. Please clarify
33) From the functional point of view, do the NKCC1 clusters in endocytic zones (EZ) differ from those outside EZ? Do they differently contribute to chloride homeostasis? Which is the relative distance of “normal” NKCC1 dendritic clusters and EZ from excitatory and inhibitory synapses?
44) The Authors state that NKCC1, unlike others proteins (e.g. GABAA receptors), do not follow the general rule by which the decrease of mobility promotes clustering while increase of mobility induces declustering. This point should be better discussed
55) The Authors use methansulphonate to substitute chloride and to lower [Cl-]I. However, by using this approach also extracellular chloride is affected: does it also interfere with NKCC1 function/structure?
66) The intracellular chloride concentration has been assessed with chloride imaging. However, electrophysiology would provide more accurate measurement of chloride reversal. Although I’m not asking for new experiment, could the Authors mention why the SuperClomeleon approach has been preferred?
Minor points:
11) Overall, I think that the manuscript would benefit from editing by a native speaker.
22) Some sentences should be more precise. Some examples are following: i) line 26, the verb “open” should relate to a channel and not a conductance, ii) line 29, “polarity” should refer to a current (or potential) and not to GABAergic transmission, iii) line 34, “depolarized” is referred to “shift” instead to “potential”, iv) line 307, the Authors should mention that the study is conducted on spontaneous activity, v) line 305: also action potential are blocked not only glutamatergic drive, vi) line 397, transfected with ?
33) Fig 1I: Intra is misspelled
Author Response
Please, see attached file.

Reviewer 2 Report
The manuscript by Come et al investigates the cellular and molecular mechanisms involved in regulation of NKCC1, a Na-K-2Cl cotransporter that participates in physiological and pathophysiological regulation of chloride homeostasis. The strategy is to visualize recombinant HA-tagged NKCC1 in hippocampal neurons using high-resolution live-imaging (Single-Particle Tracking) combined with various pharmacological and genetic manipulations affecting GABAergic activity and the WNK signaling pathway. Surface expression and clustering of NKCC1 is analysed in parallel, using fluorescence imaging and STORM microscopy in fixed samples. The results show that GABAergic activity controls the membrane diffusion and clustering of NKCC1 via the WNK1 kinase signaling pathway that targets threonine residues on NKCC1. Based on chloride imaging experiments, this pathway has no significant effects on chloride regulation in adult hippocampal neurons, but might be activated under pathological hyperactivity.
Overall, the experiments are carefully done and adequately presented. The results advance our understanding of the mechanisms regulating NKCC1 in neurons, which might be relevant for certain pathologies. The main weakness is that most of the results rely on overexpression of recombinant HA-tagged NKCC1 and no data on endogenous NKCC1 is included. Some additional minor comments are pointed out below.
1. Is NKCC1 endogenously expressed in the hippocampal cultures ? what happens to the subcellular localization and surface expression of endogenous NKCC1 upon the key manipulations used (low intracellular chloride, inhibition of the WNK1 pathway)?
2. How were axons identified (data in supplementary fig 1) ?
3. Pooled data for the diffusion coefficient and area (in the figures 1, 3, 4B, 5J, 6A) are shown on a log scale, where differences between the groups are difficult to see.
4. What is the intracellular chloride level in the hippocampal neurons before and after substituting extracellular Cl− with methane sulfonate ? The authors could use SuperClomeleon to quantify the changes in intracellular chloride concentration during this manipulation. Does the treatment influence osmolarity and/or cell volume?
Author Response
Please, see attached file.
